# Directly recruited GATA6 + peritoneal cavity macrophages contribute to the repair of intestinal serosal injury

Masaki Honda [1,3✉], Masashi Kadohisa[1,3], Daiki Yoshii[1,2], Yoshihiro Komohara[2] & Taizo Hibi[1]

Recruitment of bone marrow derived monocytes via bloodstream and their subsequent conversion to CX3CR1[+] macrophages in response to intestinal injury is dependent on CCR2, Nr4a1, and the microbiome. This process is critical for proper tissue repair; however, GATA6[+] peritoneal cavity macrophages might represent an alternative, more readily available source of mature and functional myeloid cells at the damaged intestinal locations. Here we show, using spinning-disk confocal microscopy, that large F4/80[hi]GATA6[+] peritoneal cavity macrophages promptly accumulate at damaged intestinal sites upon intestinal thermal injury and upon dextran sodium sulfate induced colitis in mice via a direct route from the peritoneal cavity. In contrast to bloodstream derived monocytes/macrophages, cavity macrophages do not depend on CCR2, Nr4a1 or the microbiome for recruitment, but rather on the ATP-release and exposed hyaluronan at the site of injury. They participate in the removal of necrotic cells, revascularization and collagen deposition and thus resolution of tissue damage. In summary, peritoneal cavity macrophages represent a rapid alternative route of intestinal tissue repair to traditional monocyte-derived macrophages.

---

[1] Department of Transplantation and Pediatric Surgery, Kumamoto University, 1-1-1 Honjo, Chuo-ku, Kumamoto 860-8556, Japan. [2] Department of Cell Pathology, Kumamoto University, 1-1-1 Honjo, Chuo-ku, Kumamoto 860-8556, Japan. [3] These authors contributed equally: Masaki Honda, Masashi Kadohisa. ✉email: honda.masaki@kuh.kumamoto-u.ac.jp

Tissue-resident macrophages are major players in maintaining local homeostasis in response to environmental changes such as infection and inflammation[1]. Unlike most other organs, the intestinal mucosal lamina propria is constantly replenished with macrophages from bone marrow progenitor monocytes[2–5]. In inflammation, rapid accumulation of Ly6C$^{hi}$ monocytes into the injured area and conversion to CX3CR1$^+$ monocytes/macrophages depends on CCR2, Nr4a1, and the microbiome[6]. This process contributes to the resolution of inflammation and tissue repair but requires at least a few days for conversion from a classical pro-inflammatory monocyte to an alternative phenotype. However, because the intestine is a portal to the external environment, we speculated that repair is initiated immediately via an alternative route during severe intestinal injury.

**Fig. 1 Large F4/80$^{hi}$ macrophages rapidly accumulate in response to intestinal thermal injury. a** Representative stitch images of colonic LP (left) in *Cx3cr1$^{GFP/+}$* mice. Representative still (middle) and three-dimensional (3D) image (right) of CX3CR1$^+$ macrophages (green) in colonic LP. Scale bars, 50 μm. **b** Representative images of colonic LP CX3CR1$^+$ monocytes/macrophages and CCR2$^+$ cells (red) 6 h after focal intestinal injury in *Cx3cr1$^{GFP/+}$Ccr2$^{RFP/+}$* mouse. Scale bars, 50 μm. **c** Migration paths, **d** crawling velocities of CCR2$^+$ cells and CX3CR1$^+$ cells in response to intestinal injury. $n = 30$/group (obtained by three independent images). **e** Quantification of large F4/80$^{hi}$ macrophages within intestinal injury at indicated time points. $n = 3$ (2 h) and 5 (6, 24, 48 h). **f** Representative image of large F4/80$^{hi}$ macrophages in injury area at 24 h post-injury. Higher magnification of the indicated area is shown in right. Scale bars, 100 μm (left) and 50 μm (right). **g** H&E and **h** immunofluorescence staining for F4/80 (brown) and GATA6 (green) of injured colon 24 h post-injury (cross section). Arrowheads indicate GATA6$^+$ peritoneal macrophages. Scale bars, 10 μm. Data were representative of three independent experiments. Data represent box and whiskers (Fig. 1d) and mean ± SEM (Fig. 1e). Source data are provided as a Source Data file.

Large GATA6$^+$ peritoneal cavity macrophages rapidly invade the subcapsular region of the liver following sterile injury/inflammation where they contribute to tissue repair[7]. The zinc finger transcription factor, GATA6, controls proliferation, survival, and metabolism[8–10]. Wang et al. revealed that peritoneal macrophages expand rapidly at liver injury sites and up-regulate CD273, CD206, and arginase-1, markers of alternatively activated macrophages[7,11,12]. The intestine is also surrounded by a peritoneal cavity; however, whether peritoneal macrophages can (1) detect luminal injury within the peritoneum, (2) invade into the intestine across a formidable mesothelial barrier under pathological conditions, and (3) affect tissue repair, have not been explored. The molecular mechanisms involved in this process are also unknown. The mammalian intestinal tract is populated with over 100 trillion microbes, the majority of which reside in the colon[13,14] so they could be a potent recruitment cue for peritoneal macrophages.

Here, we examine the recruitment and function of peritoneal cavity macrophages in intestinal injury. Using spinning-disc dual-laser intravital microscopy and a sterile burn injury model of the intestine, we detect large F4/80$^{hi}$ peritoneal macrophages accumulating at the intestinal injury site. They are recruited to the injured intestine via a unique peritoneal route in response to ATP released by necrotic cells. Interaction between hyaluronan and CD44 is indispensable for the recruitment, while CCR2, Nr4a1, and the microbiome are not involved. Accumulated large peritoneal macrophages contribute to the removal of necrotic cells within the intestinal injury site and help with revascularization and collagen deposition. We also focus on the most common form of intestinal disease, inflammatory bowel disease (IBD), and observe more severe dextran sodium sulfate (DSS)-induced colitis activity in the absence of peritoneal macrophages. Overall, these data demonstrate the importance of the immune player "GATA6$^+$ peritoneal cavity macrophages" in intestinal injury. These findings prompt further studies to explore the mechanisms and/or therapeutic strategies for disorders of the intestinal tract with a particular focus on IBD.

## Results

**Large F4/80$^{hi}$ macrophages accumulate promptly in response to sterile intestinal injury.** Intestinal macrophages express a specific cell surface marker, CX3CR1. Using intravital imaging and mice with a fluorescent reporter for CX3CR1, we imaged steady state intestinal lamina propria macrophages and found that these cells form an interdigitated physical chain surrounding the blood vessels (Fig. 1a). We examined the recruitment of monocytes and macrophages in response to sterile thermal injury of the intestine using *Cx3cr1$^{GFP/+}$Ccr2$^{RFP/+}$* mice. A 500 μm focal necrotic lesion extending into the lamina propria was created from the serosa side of the colon using a thermal probe. This model allowed us to image recruitment of immune cells in an area eradicated of resident cells. Imaging showed that CCR2$^+$ monocytes but not CX3CR1$^+$ monocytes infiltrated into the injury site within 6 h. Meanwhile, CX3CR1$^+$ macrophages that

were adjacent to the injury site remained sessile and did not move from their original position towards the damaged site (Fig. 1b–d and Supplementary Movie 1). Despite their sessile nature, when topical F4/80 antibody was applied to the injury site, a very significant population of large F4/80$^{hi}$ macrophages accumulated within 2 h post-injury in C57BL/6 mice (Fig. 1e and Supplementary Fig. 1a). The accumulation of these large F4/80$^{hi}$ cells peaked at 24 h after injury and persisted for at least 48 h (Fig. 1e and Supplementary Fig. 1a). To confirm that they were not derived from monocytes, we imaged *Cx3cr1$^{GFP/+}$Ccr2$^{RFP/+}$* mice 6 h after injury with topical administration of F4/80 antibody. This revealed that neither CCR2$^{RFP}$ nor CX3CR1$^{GFP}$ co-localized with large F4/80$^{hi}$ cells (Supplementary Fig. 1b). The neutrophil marker, Ly6G, also did not show any co-localization with F4/80 (Supplementary Fig. 1b). At 24 h after injury, CCR2$^+$ monocytes formed a ring surrounding the injury site and their accumulation was via blood vessels, whereas the large F4/80$^{hi}$ cells could not be seen in blood vessels and were positioned within the center of the injury as a large aggregate (Fig. 1f). There was a striking difference in size between CCR2$^+$CX3CR1$^+$ cells and large F4/80$^{hi}$ cells, the latter being at least twice the size of the former (Supplementary Fig. 1c, d). Importantly, these large F4/80$^{hi}$ cells in the intestinal injury site expressed GATA6, a transcription factor specific for large peritoneal cavity macrophages, but not intestinal F4/80$^+$ macrophages (Fig. 1g, h and Supplementary Fig. 2a–d). Using flow cytometry, we confirmed the intravital microscopy data showing that there was a population of GATA6$^+$ CD11b$^{hi}$F4/80$^{hi}$ macrophages in the injured colon (Supplementary Fig. 3a–c). The other populations of macrophages did not express GATA6 (Supplementary Fig. 3c).

**Peritoneal macrophages from the peritoneal cavity accumulate at the site of intestinal injury independent of CCR2 or Nr4a1.** Luminex assays revealed induction of MCP-1 (CCL2), the key chemokine that attracts CCR2$^+$ monocytes, and neutrophil chemokine, KC, in the colon at 24 h post-injury (Fig. 2a). However, the lack of the CCL2 receptor, CCR2, did not affect the recruitment of GATA6$^+$ macrophages (Fig. 2b, c) indicating that (1) CCL2 was not responsible for peritoneal macrophage recruitment and (2) CCR2-positive monocytes were not important for peritoneal macrophage recruitment. The recruitment of large F4/80$^{hi}$ cells within damaged intestine at 24 and 48 h after injury was not different in *Nr4a1$^{-/-}$* mice, which lack Ly6C$^{lo}$ monocytes at the injury site, compared with wild-type mice (Fig. 2b, c). The CX3CR1 ligand is involved in the recruitment of macrophages in some tissues; however, CX3CR1-deficient mice accumulated equivalent numbers of large F4/80$^{hi}$ macrophages to wild-type mice after intestinal injury (Supplementary Fig. 4a, b).

To confirm that the peritoneum was the source of the large F4/80$^{hi}$GATA6$^+$ macrophages, we depleted peritoneal macrophages by intraperitoneal administration of clodronate liposome (CLL), as described previously[7]. Intraperitoneal CLL treatment did not affect the distribution of intestinal CX3CR1$^+$ macrophages (Fig. 2d, e) or the accumulation of CCR2$^+$ monocytes within

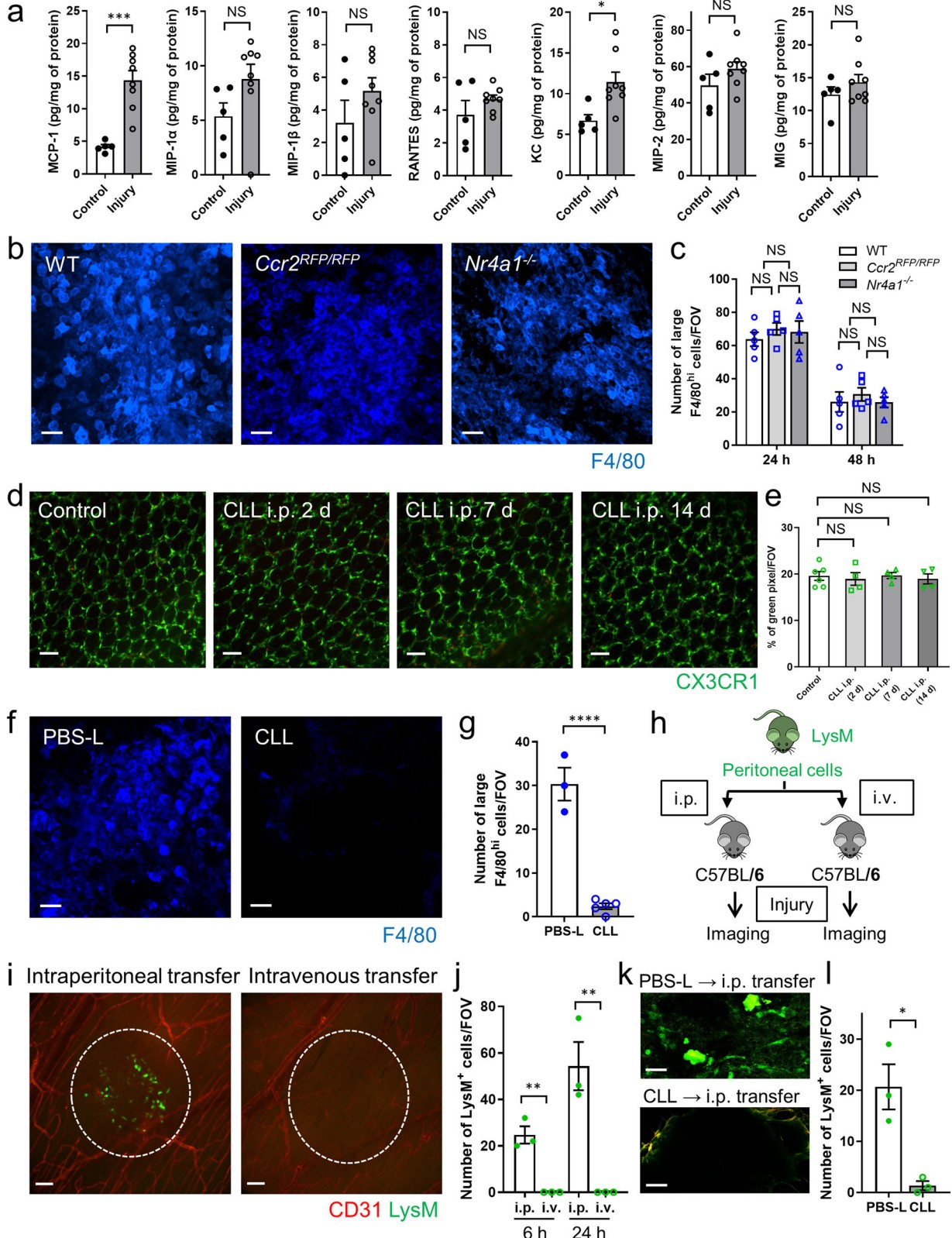

the intestinal injury site (Supplementary Fig. 5a, b) but significantly suppressed the number of large F4/80[hi] macrophages in the intestinal injury site (Fig. 2f, g). Additionally, peritoneal cells transferred from LysM-eGFP mice (in which > 85% of the GFP[+] cells in the peritoneal cavity are GATA6[+] macrophages[7]) into the peritoneum of C57BL/6 mice, resulted in accumulation of these cells at the intestinal injury site. GFP[+] macrophages were

incapable of entering the injury site when transferred intravenously (Fig. 2h–j) further indicating that this was not the route of infiltration. When peritoneal cells from LysM-eGFP mice with depleted GATA6[+] macrophages by intraperitoneal CLL administration were transferred into C57BL/6 mice intraperitoneally, GFP[+] cells were not found within the intestinal injury site (Fig. 2k, i). Next, peritoneal cells from LysM-eGFP mice were

**Fig. 2 Peritoneal Macrophages accumulate into the intestinal injury site directly via the peritoneal route regardless of CCR2 or Nr4a1. a** Luminex assays of chemokines in colon tissue samples at steady state and 24 h after thermal injury. $n = 5$ (control group) and 8 (injury group). **b** Representative images of large F4/80$^{hi}$ cells accumulated in injury site at 48 h after focal intestinal injury in WT, $Ccr2^{RFP/RFP}$, and $Nr4a1^{-/-}$ mouse. Scale bars, 50 μm. **c** The number of large F4/80$^{hi}$ cells at indicated time points are quantified ($n = 5$/group). **d** Representative images and **e** quantification of CX3CR1$^+$ cells per field of view (FOV) in lamina propria of the colon in control and 2, 7, 14 days after the intraperitoneal administration of CLL. Data were collected by imaging of $CX3CR1^{GFP/+}$ mice. $n = 4$ (CLL 2, 7, 14 d) and 6 (control). Scale bars, 50 μm. **f** Representative images of large F4/80$^{hi}$ cells accumulated in intestinal injury site at 48 h post-injury. Mice were treated by PBS-L or CLL 4 days before the injury. Scale bars, 50 μm. **g** The number of large F4/80$^{hi}$ cells are quantified. $n = 3$(PBS-L group) and 5(CLL group). **h** Schematic protocol for peritoneal cell (PC) transfer experiments (from LysM-eGFP mice to C57BL/6 mice). **i** Representative images and **j** the number of LysM-eGFP cells accumulated in C57BL/6 mice ($n = 3$/group). Scale bars, 100 μm. **k** Representative images and **l** quantification of LysM$^+$ cells within the injured colon (6 h) of C57BL/6 mice that were transferred PCs intraperitoneally from LysM-eGFP mice with i.p. administration of PBS-L or CLL. Scale bars, 50 μm. $n = 3$/group. Data represent mean ± SEM. *$p < 0.05$, **$p < 0.01$, ***$p < 0.001$, ****$p < 0.0001$, NS not significant. $P$ values were calculated with a two-tailed unpaired Student $t$-test (**a**, **g**, **j**, **l**) and one-way ANOVA followed by Tukey's post hoc test (**c**, **e**). Source data are provided as a Source Data file.

stained using a PE-ICAM2 (CD102) antibody. In total, 78.6% of GFP$^+$ peritoneal cells were also positive for CD102, which demonstrated that the double-positive cells are large peritoneal macrophages (Supplementary Fig. 6a–c). We next transferred peritoneal cells from LysM-eGFP mice to the peritoneal cavity of C57BL/6 mice after staining CD102. Imaging of the intestine at 6 h after injury showed that LysM$^+$CD102$^+$ cells had accumulated in the injured area but not in the uninjured area (Supplementary Fig. 6d–f).

**Recruitment of peritoneal cavity macrophages to the intestinal injury site is dependent on ATP and hyaluronan-CD44 interaction.** ATP is an important damage-associated molecular pattern (DAMP) in sterile injury that recruits both neutrophils and macrophages to the injury site[7]. Pretreatment with apyrase, an ATPase, or an ATP receptor antagonist inhibited the accumulation of large F4/80$^{hi}$ macrophages at the intestinal injury site (Fig. 3a, b). To further explore the mechanism of peritoneal macrophage dynamics in intestinal injury, we blocked CD44, a leukocyte adhesion molecule highly expressed on GATA6$^+$ peritoneal macrophages, and imaged intestinal injury sites. Pretreatment with anti-CD44 antibody prevented the recruitment of large F4/80$^{hi}$ macrophages to the intestinal injury site (Fig. 3c, d). Importantly, immunofluorescent staining showed exposed hyaluronan, which is the ligand for CD44, within the intestinal injury site but not on the serosal surface (Fig. 3e). Administration of hyaluronidase, an enzyme that breaks down hyaluronan, also prevented the recruitment of large peritoneal macrophages (Fig. 3f). Intriguingly, injury from the mucosal side induced some accumulation of large F4/80$^{hi}$ macrophages from the serosal surface, indicating that these accumulation mechanisms are also partially functional in severe mucosal injury (Fig. 3g, h).

**The gut microbiota is not critical for the recruitment of GATA6$^+$ peritoneal macrophages in response to intestinal injury.** Recruitment of CCR2$^+$ monocytes to the intestine and conversion to CX3CR1$^+$ monocytes/macrophages is microbiome-dependent in both the emergency repair as well as in steady state turn over[4,6,15]. Because the burn injury extended from the serosa all the way through to the lamina propria, we predicted that gut microbiota would be critical for the recruitment of GATA6$^+$ peritoneal cavity macrophages. We used broad-spectrum antibiotics from before birth until adulthood (Fig. 4a), which decimated the intestinal microbiota as assessed by SYTOX green staining (Fig. 4b). Moreover, sequencing analyses of feces showed the remarkable differences of microbial communities between control and Abx-treated mice (Fig. 4c and Supplementary Fig. 7a–d). However, these changes had no impact on the number of CD11b$^{hi}$F4/80$^{hi}$ GATA6$^+$ peritoneal macrophages found in

the peritoneal cavity nor did it affect the amount of large F4/80$^{hi}$ GATA6$^+$ macrophages that accumulated at the intestinal injury site (Fig. 4c–f), indicating that the phenotype and recruitment of large peritoneal macrophages in intestinal injury is independent of the microbiome. CD44, the predominant adhesion molecule, was also not significantly affected by the diminution of the gut microbiota (Fig. 4c, d).

**Peritoneal macrophages promote intestinal repair.** We previously reported that healing in $Ccr2^{RFP/RFP}$ mice at 48 h after injury and later was delayed. Time-lapse imaging of the colon at 24 h after thermal injury showed that large F4/80$^{hi}$ macrophages were already at the site disassembling the nearby SYTOX$^+$ necrotic cells (Fig. 5a). Next, we imaged the SYTOX green-positive cells within the intestinal injury site after depletion of peritoneal macrophages by intraperitoneal CLL administration. At 48 h post-injury, the clearance of necrotic cells was delayed in CLL-treated mice compared with PBS-liposome (PBS-L)-treated mice (Fig. 5b, c). Furthermore, peritoneal macrophage depletion was associated with significantly impaired revascularization and collagen deposition within the intestinal injury site (Fig. 5d–g). The lack of clearance of SYTOX-positive cells was even more pronounced in peritoneal macrophage-depleted $Ccr2^{RFP/RFP}$ mice (Fig. 5b, c), indicating that both CCR2$^+$ monocytes, which became CX3CR1$^+$ monocytes/macrophages, and large GATA6$^+$ peritoneal macrophages contribute to necrotic cell clearance, perhaps in separate layers of the intestine. Indeed, imaging of the injured colon showed that CCR2$^+$ monocytes accumulated mainly in the lamina propria, whereas large F4/80$^{hi}$ macrophages accumulated mainly in the muscularis (Supplementary Fig. 7a, b). The pathological analysis confirmed that the majority of F4/80$^{hi}$GATA6$^+$ macrophages were infiltrating the muscularis in the injured colon (Supplementary Fig. 5c, d). It is also worth noting that the muscularis has a lower vascular density than the lamina propria (Supplementary Fig. 6a, b), further emphasizing the importance of blood flow-independent peritoneal macrophage accumulation when the injury reaches the muscularis.

**Peritoneal macrophages accumulate in the colon in response to DSS-induced colitis.** To evaluate whether peritoneal macrophages can respond to inflammation that starts in the lamina propria and develops towards the serosa, mice were orally administered 4% DSS-containing water for 5 days. Intravital imaging of the colon in LysM-eGFP mice with F4/80 antibody topically applied to the serosa showed that LysM$^+$F4/80$^{hi}$ large macrophages were infiltrating the muscularis (Fig. 6a, b). When peritoneal cells from LysM-eGFP mice were intraperitoneally transferred to C57BL/6 mice, a similar accumulation of peritoneal LysM$^+$F4/80$^{hi}$ macrophages was found in the DSS-induced colitis

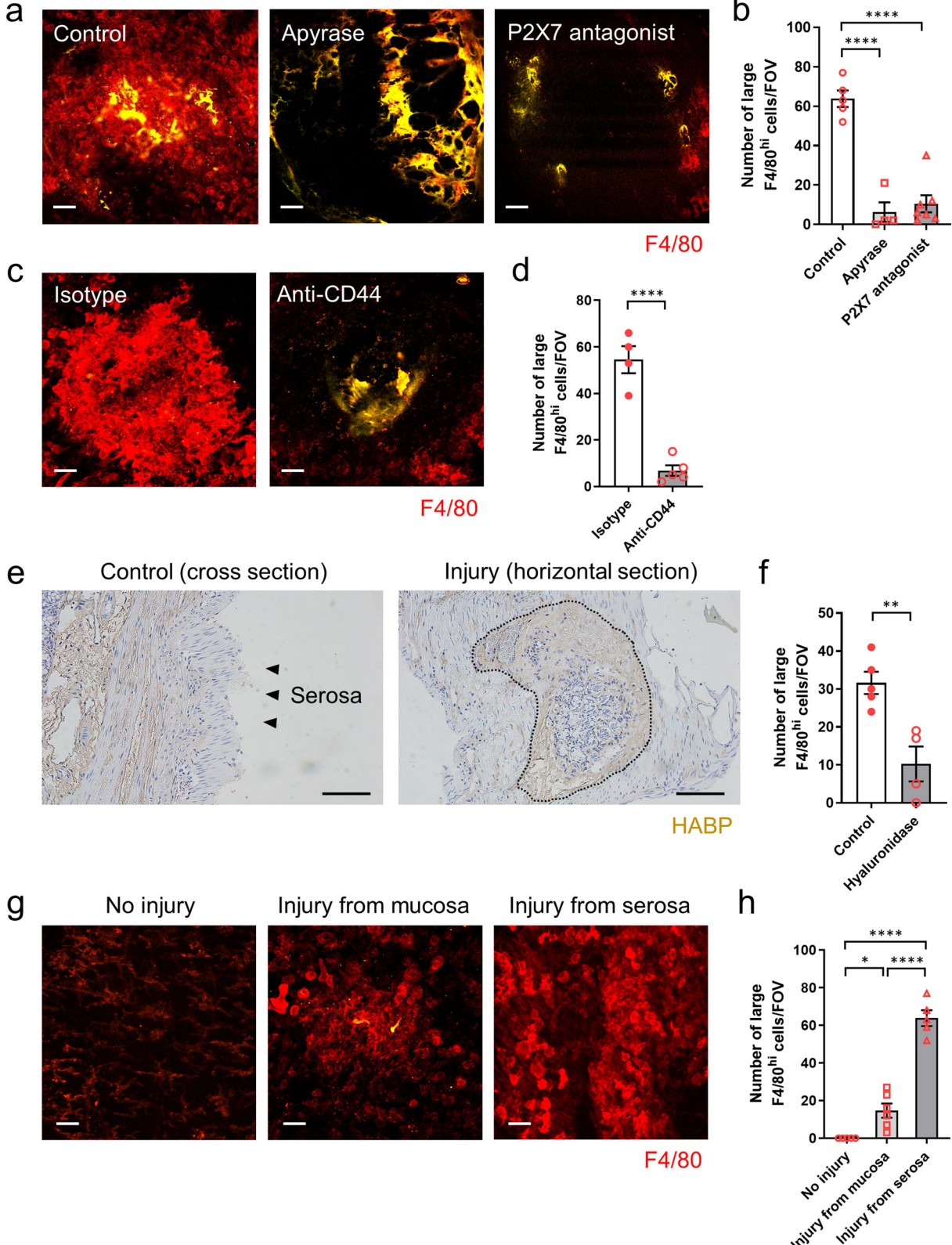

model (Fig. 6c–e). We also subjected $Cx3cr1^{GFP/+}Ccr2^{RFP/+}$ mice to DSS-induced colitis and found that no CCR2[+] and/or CX3CR1[+] signal co-localized with the large F4/80[hi] cells (Fig. 6f) indicating that these are distinct cell lineages. These findings indicate that these large invading macrophages are derived from the peritoneum, not the vasculature. For that matter, immuno-fluorescence staining showed increased expression of hyaluronan

in the colon of DSS-induced colitis compared with the control (Fig. 6g, h). As described in the thermal injury model, Abx treatment did not influence on the recruitment of F4/80[hi] large peritoneal macrophages to the colon in DSS-induced colitis (Fig. 6i, j).

To further analyze the role of peritoneal macrophages in DSS-induced colitis, we depleted peritoneal macrophages by

**Fig. 3 Recruitment of peritoneal macrophages to the site of intestinal injury depends on ATP and hyaluronan-CD44 interaction. a** Representative images and **b** quantification of the number of F4/80[hi] macrophages within intestinal injury site 24 h post-injury in mice that were pretreated with apyrase or p2rx7 antagonist. Scale bars, 50 µm. $n = 4$ (Apyrase), 5 (control), and 7 (P2X7 antagonist). **c** Representative images and **d** quantification of the number of F4/80[hi] macrophages within intestinal injury site 24 h post-injury in mice that were pretreated with isotype control or anti-CD44 antibody. Scale bars, 50 µm. $n = 4$ (isotype) and 5 (anti-CD44). **e** Immunofluorescence staining of the colon harvested from control or 2 h after injury with hyaluronic acid binding protein (HABP). The dashed line indicates the injury border. **f** Quantification of the number of F4/80[hi] macrophages within intestinal injury site 6 h post-injury in mice that were treated with hyaluronidase. Scale bars, 50 µm. $n = 4$ (hyaluronidase) and 5 (control). **g** Representative images and **h** quantification of the number of F4/80[hi] macrophages in the colonic muscularis with no injury or 24 h post-injury from mucosa or serosa. Scale bars, 50 µm. $n = 5$ (no injury, injury from serosa) and 6 (injury from mucosa). Data represent mean ± SEM. *$p < 0.05$, **$p < 0.01$, ****$p < 0.0001$. $P$ values were calculated with two-tailed unpaired Student $t$-test (**d**, **f**) and one-way ANOVA followed by Tukey's post hoc test (**b**, **h**). Source data are provided as a Source Data file.

intraperitoneal administration of CLL and compared body weight change, disease activity index, colon length, and pathological findings with those of PBS-L-administered DSS-treated mice. We found that CLL-treated mice had increased levels of injury and inflammation, including more severe weight loss and increased disease activity score (Fig. 7a, b and Supplementary Table 1). In the CLL-treated mice, colon length, which is a marker of inflammatory injury, was often shorter on the 5th day of DSS-induced colitis, but this difference was not significant compared with that of the PBS-L-treated mice (Fig. 7c). Moreover, colon tissue sections obtained 5 days after the start of DSS administration were carefully evaluated. CLL-treated mice showed more severe histomorphological scores, including marked inflammatory cell infiltration and disrupted intestinal architecture, compared with PBS-L mice (Fig. 7d, e). The degrees of infiltration by neutrophils, macrophages, and lymphocytes were compared between PBS-L and CLL groups, but no significant difference was observed in the lamina propria or muscularis, respectively (Supplementary Fig. 11a, b). Although the number of neutrophils in the muscularis of the DSS-induced colitis group treated with CLL was approximately twice that of the group treated with PBS-L, it did not reach statistical significance. Indeed, chemokine expression in the colon tissue was not affected by CLL treatment (Supplementary Fig. 11c). In the final series of experiments, peritoneal cell transfer experiments were performed for DSS-induced colitis to evaluate its effect on inflammation and symptoms (Fig. 7f). In comparison with the PBS-treated control group, the peritoneal cell transfer group had a lower disease activity index, while the other parameters did not show statistically significant differences (Fig. 7g–k).

## Discussion

The intestinal tract is a barrier to trillions of bacteria; therefore, tissue repair is vital. Resident macrophages play a pivotal role in intestinal repair helping remove debris and also helping to induce angiogenesis. However, tissue-resident CX3CR1[+] macrophages do not move during intestinal injury and depend on classical CCR2[+] monocytes to be recruited and to convert to CX3CR1[+] monocytes/macrophages. This conversion process takes more than 2 days and is dependent on the transcription factor, Nr4a1[6], indicating that early healing must be dependent on the Ly6C[hi] monocytes. However, the timing and inherent inflammatory nature of these cells make it difficult to explain how healing could begin so rapidly. In this study, we identified a third macrophage type that contributed to tissue repair, namely GATA6[+] peritoneal cavity macrophages. They rapidly accumulated via the peritoneal route at the injured site in response to intestinal injury within the serosa. Both the alarmin ATP and CD44-hyaluronan interactions were key for recruitment from the peritoneal cavity to the damaged intestinal serosa, which led to tissue repair independent of CCR2[+] monocyte-derived CX3CR1[+] monocytes/macrophages

(Fig. 8). However, somewhat surprising was the observation that these cells could also accumulate in the intestinal tract when the injury was induced via DSS colitis, an injury that starts primarily in the lamina propria.

Pleural, pericardial, and peritoneal cavity macrophages have been studied extensively, primarily in vitro[16]. The homeostatic and transcriptomic signatures of these tissue-resident macrophages are maintained by $Wt1^+$ mesothelial and fibroblastic stromal cells via the generation of retinoic acid[17]. Cavity macrophages express CD11b, F4/80, and CSF1R and can be divided into large GATA6[+] F4/80[hi] and small GATA6[-] F4/80[lo] macrophages[8–10,18–20]. Small macrophages play a pivotal role in inflammatory responses and infections[18,21]. Kim et al. showed that they are continuously replenished by blood monocytes dependent on the transcription factor, IRF4, and signals from the microbiome[22]. By contrast, the large macrophages are derived from the yolk sac and have multiple functions regarding recognition and phagocytosis of pathogens, antigen presentation, and resolution of inflammation[16]. They are also known to disappear in the early stage of infection or inflammation[21,23,24] although this may be caused by their aggregatory properties that lead to clump formation rather than actual disappearance[25]. These large GATA6[+] peritoneal macrophages express phagocytic receptors such as Tim4, MerTK, CD36, and genes for cell adhesion and angiogenesis, suggesting a repair phenotype[19,26,27]. Using intravital imaging, we visualized peritoneal GATA6[+] macrophages recruited to the injured intestine. These cells exerted an in vivo reparative function to a focal injury that started in the serosa and to DSS which causes inflammation primarily starting in the lamina propria and moving towards the serosa.

The interaction between gut microbiota and the immune system in tissue repair is an important issue. Gut microbiota can shape intestinal mucosal T cells, including Foxp3[+] regulatory T cells and mucosal-associated invariant T cells[28–30]. Additionally, Furusawa et al. provided the mechanistic insight that microbe-derived butyrate regulates the differentiation of regulatory T cells[31]. Using intravital imaging, we recently revealed that prenatal antibiotic administration and germ-free conditions altered the localization of intestinal mucosal CX3CR1[+] macrophages and impeded the development of a perivascular anatomical barrier[6]. By contrast, the microbiome did not regulate the function of GATA6[+] peritoneal cavity macrophages because their numbers were not altered in mice given antibiotics from birth and they still accumulated the injured intestine. Despite significant alterations in many different immune cells in the dysbiotic intestine, none of the many differences affected peritoneal macrophage accumulation to the injured intestine indicating that these cells are not affected by alterations in the microbiome or intestinal immunobiome.

This study also has clinical implications because humans have an intraperitoneal mature CD14[hi]CD16[hi] macrophage subset that shows intracellular GATA6 expression[32]. As a typical digestive

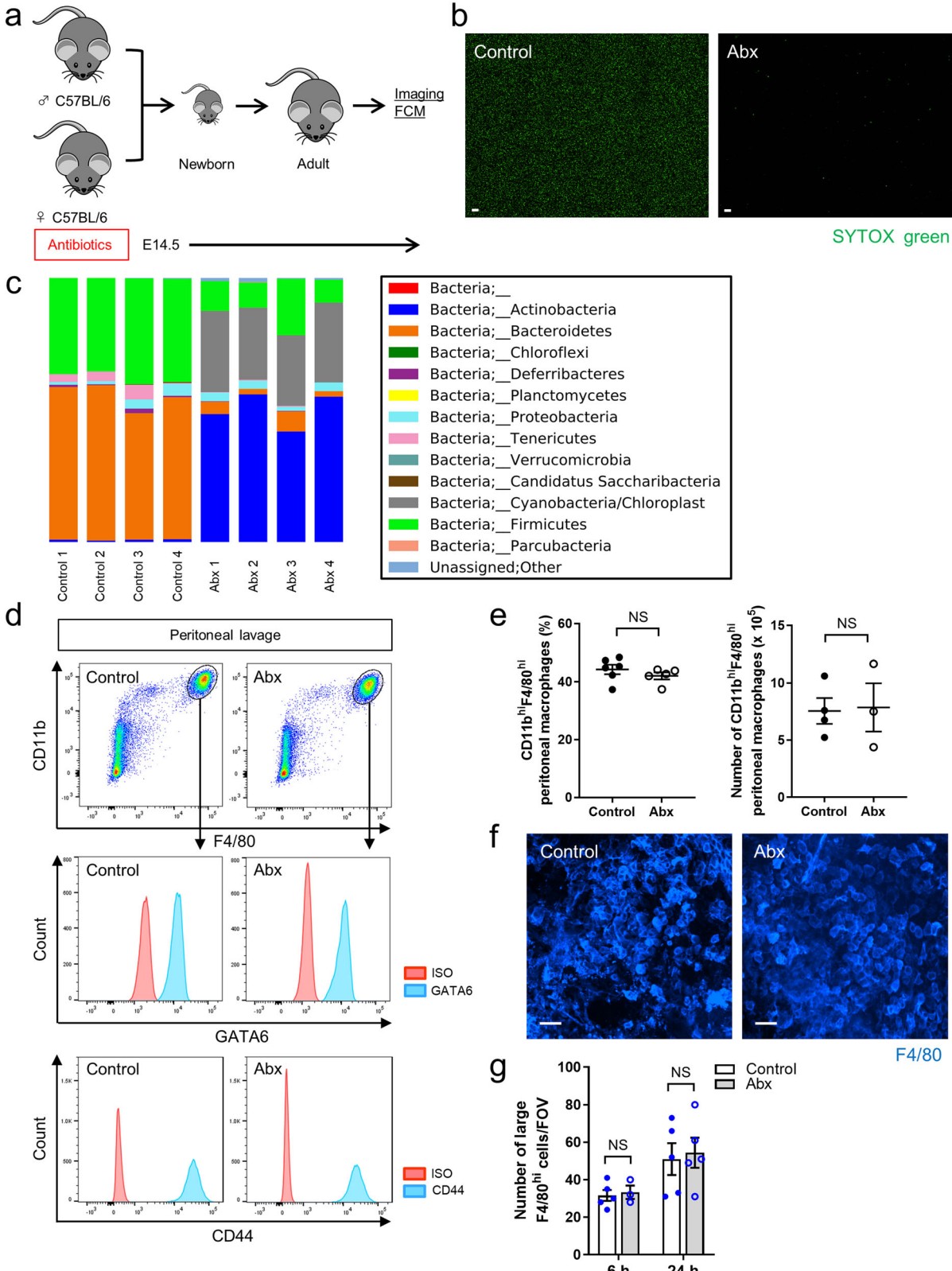

SYTOX green

GATA6

CD44

F4/80

tract disease, IBDs, including ulcerative colitis and Crohn's disease, are a worldwide burden with high incidence in developed countries and increasing incidence in developing countries[33]. Currently, new therapeutic strategies targeting the recruitment and activation of immune cells from the vasculature are being tested[34]. Our results raise the intriguing possibility that enhancing

GATA6[+] macrophage function is also worth considering as a form of therapy. Indeed, adoptive transfer of peritoneal macrophages with or without IL33, which can induce the proliferation and alternative activation of peritoneal macrophages, ameliorated inflammation in a preclinical colitis model[35,36]. Whether these peritoneal cavity macrophages also contribute to colorectal cancer

**Fig. 4 Gut microbiota does not affect the phenotype and dynamics of peritoneal macrophages in intestinal injury. a** Schematic protocol for the generation of dysbiotic mice. **b** Representative stitch images (from 12 different fields of view) of SYTOX green[+] particles in intestinal contents of SPF control and Abx-treated mice. Scale bars, 50 μm. **c** Relative abundance of bacterial amplicon sequence variants in feces obtained from control or Abx-treated mice. The bar plot is displayed at the phylum level. $n = 4$/group. **d** Flow cytometry analysis of GATA6 and CD44 expression in peritoneal CD11b[hi]F4/80[hi] macrophages obtained from control or Abx-treated mice. Cells were pregated on size, viability, and CD45[+]. Data were representative of three independent experiments. **e** Quantification of the proportion of peritoneal CD11b[hi]F4/80[hi] macrophages (left). $n = 6$ (control) and 5 (Abx). An absolute number of peritoneal CD11b[hi]F4/80[hi] macrophages (right). $n = 4$ (control) and 3 (Abx). **f** Representative images of the large F4/80[hi] macrophages within intestinal injury in control and Abx mice. Scale bars, 50 μm. **g** Quantification of the number of large F4/80[hi] macrophages in injury site at 6 and 24 h post-injury. $n = 3$ (Abx at 6 h) and 5 (control at 6 h, control and Abx at 24 h). Data represent mean ± SEM. NS not significant. P values were calculated with a two-tailed unpaired Student t-test (**e**, **g**). Source data are provided as a Source Data file.

and other intestinal diseases is an intriguing but as yet untested hypothesis. Further studies are needed to address this question.

In summary, this study demonstrates GATA6[+] peritoneal macrophages to be mobile immune players with a specific repair function in intestinal injury and inflammation. Our data indicate that enhancing their repair functions is a potential therapeutic approach for numerous intestinal diseases that include IBD.

## Methods

**Mice.** C57BL/6 mice (#000664), LysM-eGFP mice, $Cx3cr1^{GFP/GFP}$ (CX3CR1-deficient) mice (#005582), and $Nr4a1^{-/-}$ mice (#006187) were obtained from The Jackson Laboratory. Generation of $Ccr2^{RFP/RFP}$ (CCR2-deficient) mice have been previously described[37]. $Cx3cr1^{GFP/+}Ccr2^{RFP/+}$ mice were generated by crossing $Cx3cr1^{GFP/GFP}Ccr2^{RFP/RFP}$ mice with C57BL/6 mice[37]. All mice were on a C57BL/6 background. Animals were maintained in a specific pathogen-free environment with ad libitum access to food and water. Mice were housed under standardized conditions of temperature (21–22 °C) and illumination (12/12 h light/dark cycle). Mice of 8–12 weeks of age were used for experiments. Mice were gender-matched for experiments and experimental/control mice were bred separately. Mice were euthanized by cervical dislocation after imaging or for tissue sampling. All experiments were approved by the Kumamoto University Ethics Review Committee for Animal Experimentation and were performed according to guidelines of the Institutional Animal Committee of Kumamoto University.

**Antibodies and reagents.** Antibodies against CD11b (#17-0112-82; M1/70) (1:100 dilution), CD31 (#12-0311-82; PECAM-1, 390) (1:100 dilution), CD45 (#45-0451-82; 30-F11) (1:150 dilution), F4/80 (#12-4801-82; BM8) (1:100 dilution) were obtained from eBioscience. Antibodies against Ly6G (#127612; 1A8) (1:100 dilution) were obtained from Biolegend. Antibody against CD44 (#553134; IM7) (1:150 dilution) was obtained from BD Biosciences. Antibody against GATA6 (#26452; D61E4) (1:50 dilution) was obtained from Cell Signaling Technology. SYTOX Green (S7020) was obtained from Thermo Fisher Scientific. For intravital imaging, we used a very low concentration at 2 μg per mouse of each antibody. Each antibody was diluted in PBS as appropriate.

**In vivo treatment.** Peritoneal macrophages were depleted by intraperitoneal administration of 100 μL/mouse (0.69 mol/L) clodronate liposome (#CP-020-020; clodronateliposomes.org, Vrije Universiteit, Netherlands) 4 days prior to the experiment. The same dose of PBS-liposome (#CP-020-020) was used for the control experiment. For adhesion molecule blocking experiments, mice were administered either 50 μg anti-CD44 (CL8944AP; KM81, Cedarlane) monoclonal antibody or isotype control antibody 1 h prior to the injury. Apyrase treatment was performed by intraperitoneal administration of 25 U apyrase (#A2230; Sigma-Aldrich). For blocking ATP receptor, mice received 10 μg P2X7 antagonist (#A438079; R&D) intraperitoneally 1 h prior to the injury. Hyaluronidase treatment was performed by intraperitoneal administration of 100 U hyaluronidase (#H3506; Sigma-Aldrich) just after the intestinal injury. Each reagent was diluted in PBS as appropriate.

**Preparation of the mouse intestine for intravital imaging.** Mice were anesthetized by s.c. injection of 200 mg/kg ketamine (Bayer Animal Health) and 10 mg/kg xylazine (Bimeda-MTC). After anesthesia, the right jugular vein was cannulated to administer fluorescent dyes and additional anesthetic. For intestinal imaging, a midline incision followed by a left lateral incision along the costal margin to the midaxillary line was performed to expose the intestine. To image the intestinal wall from the serosal site, the oral side of the intestine was ligated with 5-0 silk string. PBS (200 μL/20 mm of the colon) was introduced into the intestinal lumen using a syringe with a 30-gauge needle, and the anal side of the intestine was ligated with 5-0 silk string. Mice was placed in a left lateral position and the intestine was placed on a glass coverslip and imaged if blood flow was normal. We defined "muscularis" as 10–30 μm of depth, "submucosa" as 30–50 μm of depth, and "lamina propria" as 50–80 μm of depth by imaging from serosa. Mice placed on a heating pad to

maintain a body temperature of 37 °C throughout imaging. Exposed abdominal tissues were covered with saline-soaked gauze to prevent dehydration.

**Spinning-disc confocal intravital microscopy (SD-IVM) and image analysis.** A multichannel spinning-disk confocal microscope was used to image the mouse intestine. Image acquisition of the intestine was performed using an Olympus IX81 inverted microscope (Olympus, Center Valley, PA), equipped with an Olympus focus drive and a motorized stage (Applied Scientific Instrumentation, Eugene, OR) and fitted with a motorized objective turret equipped with 4×/0.16 UPLAN-SAPO, and 10×/0.40 UPLANSAPO objective lenses and coupled to a confocal light path (WaveFx; Quorum Technologies) based on a modified CSU-10 head (Yokogawa Electric Corporation, Tokyo, Japan). Cells of interest were visualized using fluorescently labeled antibodies, and fluorescent reporter mice. In some experiments, necrotic cells were labeled by superfusion of the intestinal surface with 1 μM SYTOX Green. Laser excitation wavelengths of 491, 561, 642, and 730 nm (Cobolt, AB, Solna, Sweeden) were used in rapid succession together with the appropriate band-pass filters (Semrock Inc., Rochester, NY). A back-thinned electron-multiplying charge-coupled device 512 × 512-pixel camera (Hamamatsu Photonics) was used for fluorescence detection. Volocity software 6.1 (PerkinElmer) was used to drive the confocal microscope and analysis of images.

Acquired images were analyzed or exported as TIF images using Volocity software. The minimum threshold values were adjusted for each of the fluorescence channels to reduce the background. Exported images were imported to the Image J software package (NIH) for analysis[38]. For quantification in the number of large F4/80[hi] macrophages in the intestine, images were acquired for each mouse using a 10× objective and was evaluated using the "analyze and measure" command in Image J software. F4/80[hi] cell whose length is larger than 15 μm was defined as "large" macrophage. Quantification of SYTOX Green-positive dead cells in the intestinal injury area was performed using Image J as previously described[7,38]. Autofluorescence induced by debris was excluded from the analysis.

**Depletion of gut commensal bacteria.** Gut commensal bacteria were depleted using a method modified a protocol as previously described[39]. Mice were provided with ampicillin (1 g/L), vancomycin (0.5 g/L), neomycin (1 g/L), metronidazole (1 g/L), and ciprofloxacin (0.2 g/L) in drinking water. All antibiotics were obtained from Sigma-Aldrich. Antibiotics-mixed water was started from E14.5 and continued until the experiments.

**Sterile inflammation induced by focal necrotic injury.** Mice were anesthetized with isoflurane and a small midline laparotomy was made to exteriorize the colon. For imaging, a single focal injury was induced on the serosal surface of the colon to a depth of 80 μm using the tip of a heated 30-gauge needle mounted on an electrocautery device. Injury from the mucosal side was performed by puncturing the needle from the opposite side, and the puncture hole was sutured closed. After the induction of thermal injury, the abdominal incision was sutured closed and mice were allowed to recover for imaging of later time points (2, 6, 24, 48 h). For sham experiments, mice underwent the same surgical procedure, but no injury was induced.

**Immunocytochemistry.** Peritoneal macrophages were obtained from peritoneal exudates of mice, and erythrolysation was performed. Peritoneal macrophages were cultured in low-glucose DMEM supplemented with 2% FBS and 1% penicillin/streptomycin. Cytospin smears were prepared by placing 1 mL ($1.0 \times 10^5$ cells/ml) cultured fluid in the cytospin funnel with filter paper being placed between the funnel and the slide, followed by centrifugation at 800 rpm for 5 min resulting in the formation of a monolayered sheet of cells within a small circumference. After drying the sections, sheeted peritoneal macrophages were fixed with 1% paraformaldehyde for 5 min. Subsequently, the sections were incubated with blocking solution (1.0% bovine serum albumin in TBS) for 20 min at room temperature (RT), then reacted with primary antibodies. The sections probed with primary antibodies were incubated with secondary antibodies. Antibodies used in the immunofluorescence (IF) method are listed in Supplementary Table 2. IF sections were mounted using a mounting medium containing 4′,6-diamidine-2′-phenylindole dihydrochloride (DAPI) (#SCR-38448; Dianova GmbH, Hamburg, Germany).

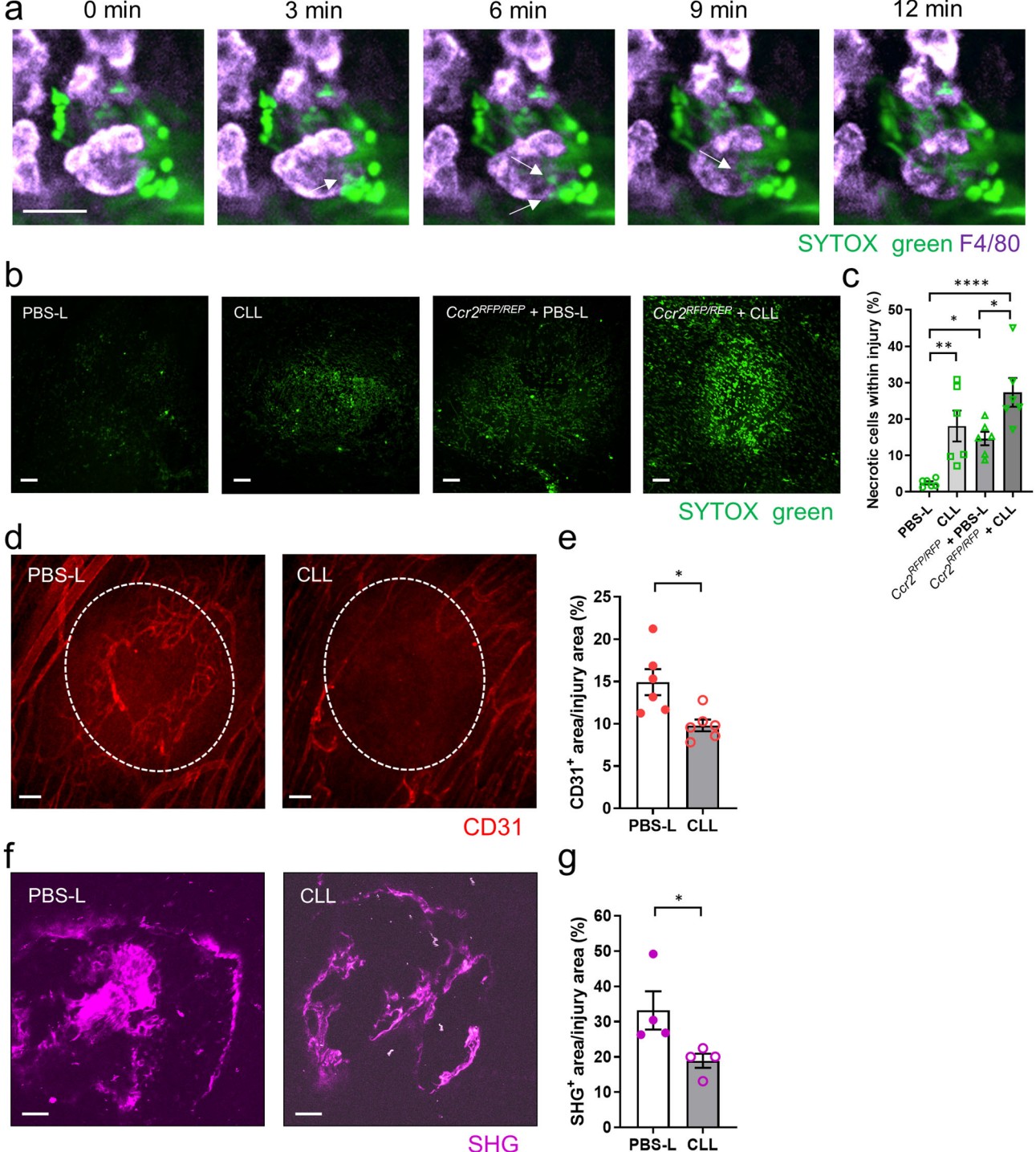

**Fig. 5 Peritoneal cavity macrophages facilitate intestinal repair. a** Time-lapse images showing F4/80hi large macrophage pulling off SYTOX green+ particles within the intestinal injury site. Elapsed time is shown. Scale bar, 20 μm. **b** Representative images of SYTOX green+ cells within intestinal injury site at 48 h post-injury in PBS-L- or CLL-treated *CCR2+/+* and *CCR2RFP/RFP* mice. Scale bars, 100 μm. **c** Quantification of SYTOX green+ area within injury in each group. $n = 6$/group. **d** Representative images of intestinal injury site 48 h after injury in PBS-L- or CLL-treated mice. Mice were administered an anti-CD31 (red) antibody intravenously to visualize vasculature. The white dashed line highlights the original injury border. Scale bars, 100 μm. **e** Quantification of revascularization (CD31+ area within injury). $n = 6$/group. **f** Representative images of collagen (purple) within intestinal injury site 72 h post-injury in PBS-L- or CLL-treated mice. Scale bars, 50 μm. **g** Quantification of collagen deposition. $n = 4$/group. Data represent mean ± SEM. *$p < 0.05$, **$p < 0.01$, ****$p < 0.0001$. *P* values were calculated with two-tailed unpaired Student *t*-test (**e**, **g**) and one-way ANOVA followed by Tukey's post hoc test (**c**). Source data are provided as a Source Data file.

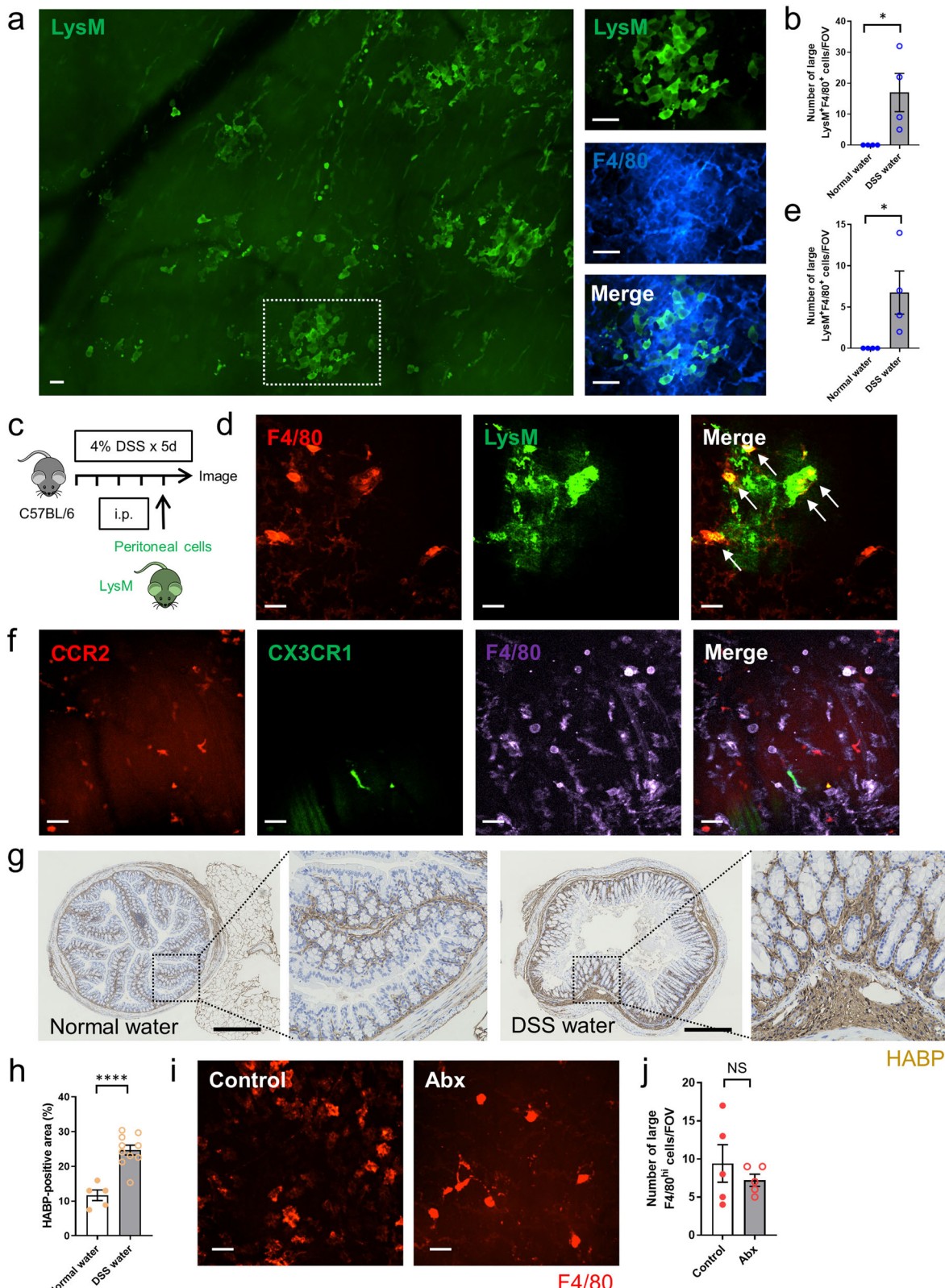

**Immunohistochemistry**. Specimens were fixed at RT for from 24 to 48 h in 10% formalin (#134-10047; FUJIFILM Wako Pure Chemical, Corp., Osaka, Japan) and embedded in paraffin (#7810; Sakura Finetek Japan Co., Ltd., Tokyo, Japan). To prepare the paraffin-embedded cell block specimens, obtained peritoneal macrophages were fixed in 10% neutral buffered formalin. Then cells were suspended in 1% sodium alginate and solidified by the addition of 1 M calcium chloride. Finally, gelatinous specimens containing macrophages were embedded in paraffin in a routine manner. Specimens were then sectioned at 3 μm. Sections were pretreated

for antigen retrieval. After cooling down to RT, the sections were treated with 0.3% $H_2O_2$ in methanol (30 min at RT) and were subsequently incubated with the blocking solution for 20 min at RT, following which the sections were reacted with primary antibodies. The sections were then incubated with horseradish peroxidase (HRP)-labeled secondary antibodies. Antibodies used in the immunoenzyme method are listed in Supplementary Table 2. Immunoreactions were visualized using the diaminobenzidine (DAB) substrate kit (#425011, Nichirei Biosciences). For double-immunostaining, sections visualized with DAB in the first

**Fig. 6 Peritoneal macrophages accumulate to the colon in DSS-induced colitis. a** Representative stitch images of the colon in LysM-eGFP mice 5 days after the start of 4% DSS-containing water. Higher magnification of the indicated area was shown in right. Scale bars, 50 μm. **b** Quantification of the number of large LysM+F4/80hi cells per FOV in LysM-eGFP mice that were treated with normal water or DSS-containing water. n = 4/group. **c** Schematic protocol for peritoneal cell transfer experiments (from LysM-eGFP mice to C57BL/6 mice). **d** Representative images of the colon in C57BL/6 mice that was administered peritoneal cells obtained from LysM-eGFP mice intraperitoneally 24 h before the imaging. Images were taken 5 days after the start of 4% DSS-containing water. Anti-F4/80 antibody (red) was applied topically to the serosa. Arrows indicate LysM+F4/80hi cells. Scale bars, 50 μm. **e** Quantification of the number of large LysM+F4/80hi cells per FOV in WT mice that were treated with normal water or DSS-containing water. Mice were pretreated by intraperitoneal administration of peritoneal cells from LysM-eGFP mice 24 h before the imaging. n = 4/group. **f** Representative images of the colon in Cx3cr1GFP/+Ccr2RFP/+ mice 5 days after the start of 4% DSS-containing water. Anti-F4/80 antibody (purple) was applied topically. Scale bars, 50 μm. Data were representative of three independent experiments. **g** Immunofluorescence staining of the colon harvested from control or 5 days after the start of 4% DSS-containing water with hyaluronic acid binding protein (HABP). Scale bars, 500 μm. **h** Quantification of the HABP-positive area (%). n = 5 (normal water) and 10 (DSS water). **i** Representative images of the colonic muscularis in control and Abx mice 5 days after the start of 4% DSS-containing water. Anti-F4/80 antibody (red) was applied topically to the serosa. Scale bars, 50 μm. **j** Quantification of the number of large F4/80hi cells per FOV. n = 5/group. Data represent mean ± SEM. *p < 0.05, ****p < 0.0001. P values were calculated with a two-tailed unpaired Student t-test (**b**, **e**, **h**, **j**). Source data are provided as a Source Data file.

immunostaining were then used to perform the second immunostaining, with the reaction visualized using HistoGreen (#E109; LINARIS Biologische Produkte GmbH, Dossenheim, Germany). Sections were mounted using malinol (Muto Pure Chemicals Co., Ltd, Tokyo, Japan).

**Image processing, cell counting, and area measurement for immunostained sections**. Immunostained sections were photographed using a microscope (BX51, Olympus Corporation, Tokyo, Japan). To quantify immunostaining of Gr-1 by cell counting, two pathologists (D.Y. and Y.K.), blinded to mouse information evaluated sections. To quantify immunostaining by calculating areas of immunostained sections, we used BZ-X800 (Keyence corp., Osaka, Japan). Immunostained sections were photographed and then entire fields were reconstructed into one picture. IBA1 and CD3 area proportions of the colonic mucosa and muscularis for each entire field were measured, respectively. HABP area proportion for the entire field was also measured.

**Peritoneal cell transfers**. Peritoneal lavage was performed as described before using sterile PBS[18]. Cells were washed with cold PBS twice and resuspended in 100 μL PBS. 5 × 10^6 cells isolated from LysM-eGFP mice were directly transferred into naïve mice. Both intraperitoneal and intravenous transfers were performed 1 h prior to injury induction. In the 4% DSS-induced colitis model, peritoneal cells isolated from LysM-eGFP mice were transferred 4 days after the start of DSS-containing water and the colon was imaged 24 h later.

**SYTOX green staining of intestinal content**. Bacterial load of intestinal contents was measured as previously described[40]. Briefly, a fecal pellet was collected from control or an Abx-treated mouse and each sample was added 500 μL 4% paraformaldehyde and mixed. After the incubation for 30 min at room temperature, 100 μL of fixed content was resuspended in 800 μL sterile PBS and 1 μL of SYTOX green (100 μg/mL) was added. After the incubation for 30 min in the dark, samples were centrifuged and 50 μL of supernatant was spread on a slide. The image was recorded by microscopy in a green fluorescence channel using a 10× objective.

**Detection of intestinal collagen by multi-photon microscopy**. Intestinal fibrillar collagen was imaged using second-harmonic generation (SHG). In brief, focal injured colons were removed, maintained in cold PBS, and imaged using a BX61WI upright microscope and FV1000MPE (Olympus, Tokyo, Japan) laser-scanning microscope system equipped with a MaiTai HP Deep See femtosecond-pulsed laser (Spectra-Physics, Santa Clara, CA, USA). Colonic tissue was visualized using a combination of multi-photon fluorescence and SHG using 890 nm excitation[41]. The signal was detected by the external non-descanned detectors (505 nm mirror and band-pass emission filters at 465–485 nm for SHG). Z-stacks were recorded at 0.5 μm intervals with an Olympus XLPLN 25X WMP (water immersion; numerical aperture, 1.05; working distance 2.0 mm) objective lens.

**Flow cytometry**. After mice were anesthetized, colon or peritoneal lavage were collected and placed in PBS on ice. Colonic cells were obtained from intestinal biopsies of the uninjured or injured area. After the homogenization, single-cell suspensions were generated by mechanical disruption through a 40-μm nylon mesh (BD Bioscience). Residual red blood cells were lysed using ACK lysing buffer (Invitrogen). The cells were blocked using anti-CD16/32 antibody (#BE0307; 2.4G2 clone; Bio X Cell) (1:100 dilution) for 30 min. Then, cells were stained for 30 min with antibodies for specified markers. Nonviable cells were identified using viability dye efluor 780 (#65-0865-14; eBioscience) (1:1000 dilution) or Ghost Dye™ Red 710 (#13-0871-T100; TONBO biosciences) (1:1000 dilution). A Foxp3 staining buffer set (eBioscience) was used for intracellular GATA6 staining. Samples were

run using a flow cytometer (FACSCanto; BD Biosciences) and analyzed using FlowJo software (Tree Star).

**Measurement of colon tissue chemokine concentration**. Quantification of concentrations of chemokines was performed using the validated Luminex bead-based assay from R&D Systems (Minneapolis, MN) according to the manufacturer's instructions. Briefly, a 20-mm-long colon tissue sample was placed into 1 mL PBS, and they were mechanically disrupted. The mixture was then centrifuged at 20,124 × g for 5 min three times and the supernatant was transferred to a 0.22 μm PVDF DuraPore centrifugal filter (EMD Millipore, Billerica, MA) to remove any particles from the solution. Filtered samples were then incubated with a capture bead cocktail on a 96-well plate in the dark for 2 h at room temperature. After incubation, wells were washed with wash buffer, incubated with a biotin antibody cocktail for 1 h next before another round washing, and incubated for another 30 min with Streptavidin-PE. Following washes, the plate was read using a Luminex 200 apparatus (Applied Cytometry Systems, UK) and analyzed with StarStation V.2.3 (Applied Cytometry Systems, UK). Total protein concentration of filtered samples was measured using Bio-Rad Protein Assay (Bio-Rad Laboratories). Results were normalized against the amount of total protein extracted from the colon tissues or the weight of the colon tissue.

**Bacterial DNA extraction and amplification**. DNA was extracted from fecal pellet using a QIAamp Fast DNA Stool Mini Kit (QIAGEN) according to manufacturer's instructions and 16 S rRNA genes were amplified and sequenced using an Illumina MiSeq (Illumina, San Diego, CA, U.S.A.) as described previously[42]. The 16 S rRNA operational taxonomic units (OTUs) were selected from the combined reads using a de novo OTU picking protocol clustered at 97% identity using the Quantitative Insights Into Microbial Ecology (QIIME) pipeline software. To investigate the bacterial diversity of each sample, the number of OTUs, Chao 1, and Shannon were calculated, and a rarefaction curve was generated using QIIME. Differences of microbial communities were evaluated using phylogeny-based unweighted or weighted UniFrac distance matrices. Principal coordinate analysis (PCoA) graph and Unweighted Pair Group Method with Arithmetic mean (UPGMA) tree graph were depicted using QIIME.

**DSS colitis model**. Mice were given 4.0% DSS (molecular mass, 36,000–50,000, MP Biomedicals, LLC) in the drinking water for continuous 5 days. Body weight and disease activity score was observed daily in PBS-L treated control and CLL-treated mice. Disease activity index was evaluated according to the previous report[43] and was described as follows; (a) general appearance: normal = 0; piloerection = 1; lethargy and piloerection = 2; motionless, sickly = 4; (b) weight loss: no change = 0; < 5% = 1; 6–10% = 2; 11–20% = 3; > 20% = 4; (c) feces consistency: normal = 0; pasty, semi-formed = 2; liquid, sticky, or unable to defecate after 5 min = 4; and (d) rectal bleeding: no blood = 0; visible blood in rectum = 2; visible blood on fur = 4. At 5 days after the start of DSS-containing water, colon length was measured, and the pathological findings were evaluated in a blinded fashion using a previously reported scoring system:[44] (a) inflammatory cell infiltrate (severity, extent): mild, mucosa = 1; moderate, mucosa and sub-mucosa = 2; marked, transmural = 3; (b) intestinal architecture (epithelial changes, mucosal architecture): focal erosions = 1; erosions ± focal ulcerations = 2; extended ulcerations ± granulation tissue ± pseudopolyps = 3.

**Statistical analysis**. Data were expressed as mean ± SEM. Unpaired Student t-test or Mann–Whitney U-test was used to compare between two groups as appropriate. One-way ANOVA was used to compare more than two groups, followed by Tukey's post hoc test. Percent body weight change or disease activity index in PBS-L- or CLL-treated mice with 4% DSS colitis were compared by two-way repeated-measures ANOVA.

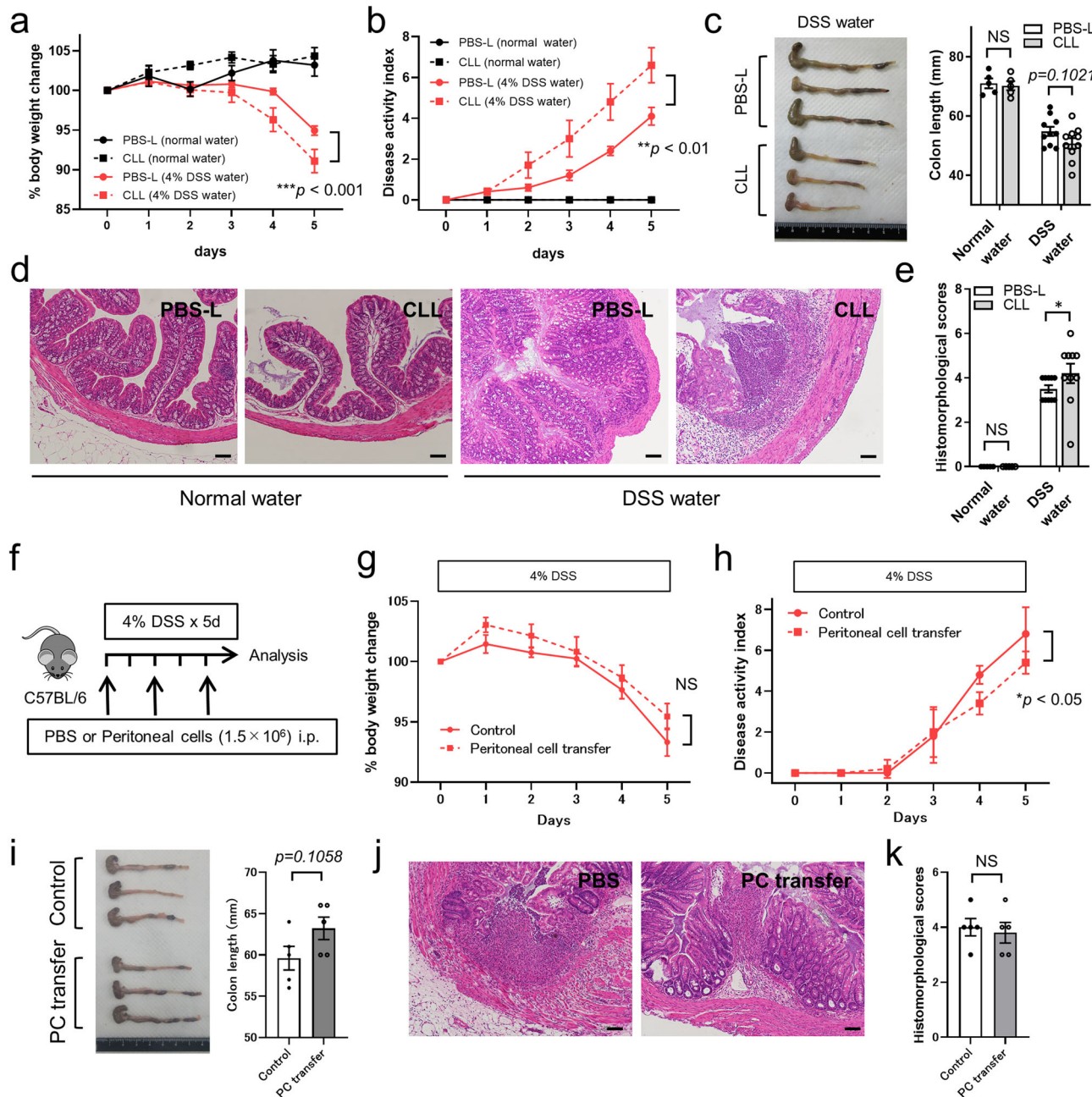

**Fig. 7 Peritoneal macrophages exert a tissue repair function in DSS-induced colitis. a** Body weight change of PBS-L treated control and CLL-treated mice in 4% DSS-induced colitis. $n = 5$ (normal water group treated by PBS-L or CLL) and 20 (DSS water group treated by PBS-L or CLL). Data were pooled from three independent experiments. **b** Disease activity index of PBS-L treated control and CLL-treated mice in 4% DSS-induced colitis. $n = 5$ (normal water group treated by PBS-L or CLL) and 10 (DSS water group treated by PBS-L or CLL). **c** Macroscopic findings of the colon 5 days after the start of 4% DSS-containing water (left) and quantification of colon length (right). $n = 5$ (normal water group treated by PBS-L or CLL) and 10 (DSS water group treated by PBS-L or CLL). **d** Representative H&E staining of the colon 5 days after the start of 4% DSS-containing water and **e** quantification of histomorphological scores. $n = 5$ (normal water group treated by PBS-L or CLL) and 10 (DSS water group treated by PBS-L or CLL). Scale bars, 100 μm. **f** Schematic protocol for peritoneal cell (PC) transfer experiments in DSS-induced colitis. **g** Body weight change and **h** disease activity index of PBS-treated control and PC-treated mice in 4% DSS-induced colitis. $n = 5$/group. **i** Macroscopic findings of the colon 5 days after the start of 4% DSS-containing water (left) and quantification of colon length (right). $n = 5$/group. **j** Representative H&E staining of the colon 5 days after the start of 4% DSS-containing water and **k** quantification of histomorphological scores. $n = 5$/group. Scale bars, 100 μm. Data represent mean ± SEM. *$p < 0.05$, **$p < 0.01$, ***$p < 0.001$. P values were calculated with two-tailed unpaired Student $t$-test (**c**, **i**), Mann–Whitney $U$-test (**e**, **k**), and two-way repeated-measures ANOVA (**a**, **b**, **g**, **h**). Source data are provided as a Source Data file.

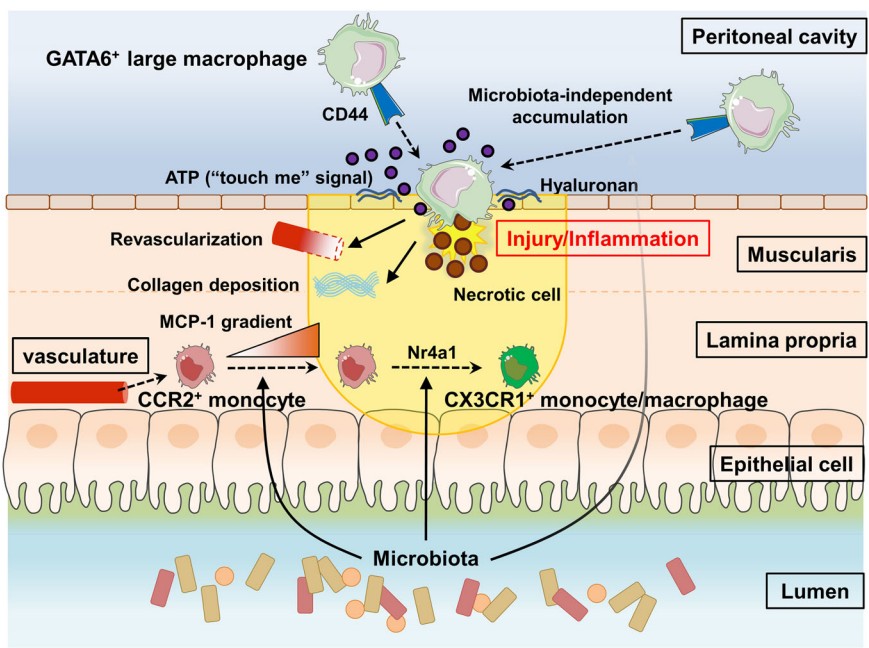

**Fig. 8 GATA6$^+$ peritoneal cavity macrophages infiltrate an injured intestine via a direct peritoneal route and contribute tissue repair.** A scheme showing the mechanism of GATA6$^+$ peritoneal macrophage recruitment and bloodstream-derived CCR2$^+$ monocyte to CX3CR1$^+$ monocyte/macrophage conversion in intestinal injury/inflammation.

Experimental findings were reproduced at least twice to ensure consistency. A $p$ value <0.05 was considered statistically significant. All tests were two-tailed. All statistical analyses were performed using GraphPad Prism v8.0 software (GraphPad Software Inc., La Jolla, CA).

**Reporting summary**. Further information on research design is available in the Nature Research Reporting Summary linked to this article.

## Data availability

The sequence data generated in this study have been deposited in the NCBI repository GenBank Nucleotide under accession numbers OK665944-OK666378. The first is accessible and the others can be accessed by editing the accession number in the hyperlink. Further data that support the findings of this study are available from the corresponding author upon reasonable request. Source data are provided with this paper.

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

## Acknowledgements

We would like to thank Servier for providing Servier Medical Art, which was used for the creation of figures (Fig. 8 and Supplementary Fig. 7a). Servier Medical Art by Servier is licensed under a Creative Commons Attribution 3.0 Unported License (https://creativecommons.org/licenses/by/3.0/). The source of the mouse icons used in Figs. 2h, 4a, 6c, 7f, and Supplementary Figs. 6a, 6d, 10a are Free vector graphics on Pixabay. Content on Pixabay is licensed under Pixabay License (https://pixabay.com/service/terms/#license). We also thank Jeremy Allen, Ph.D., from Edanz Group (https://en-author-services.edanzgroup.com/ac) for editing a draft of this manuscript. M.H. is supported by the Research Fellowship of the Uehara Memorial Foundation and grants from the Ministry of Education, Culture, Sports, Sciences and Technology of Japan (KAKENHI 19H03716).

## Author contributions

M.H. designed and performed experiments, analyzed data, and wrote the manuscript. M.K. performed experiments, analyzed data, and wrote the manuscript. D.Y. performed experiments and analyzed pathological specimens. Y.K. and T.H. provided material support and reviewed the manuscript.

## Competing interests

The authors declare no competing interests.
