## [Peer Review File · Nature Communications]

Directly recruited GATA6+ peritoneal cavity macrophages contribute to the repair of intestinal serosal injuryREVIEWER COMMENTS

Reviewer #1 (Remarks to the Author):

This study describes a novel concept in the intestinal repair arena, i.e., peritoneal macrophages appear to be important in helping repair the intestinal muscularis layer. I have a number of queries.

- 1) Would DSS colitis cause greater injury in CCR2^{-/-} mice if peritoneal macrophages are depleted?
- 2) What would happen if the burn injury occurred from the luminal side of the intestine without breaching the serosal layer? Would the GATA6⁺ macrophages still accumulate or would this be invisible to the peritoneal macrophages?
- 3) The DSS colitis shows a remarkable increase in immune cells throughout the intestinal layers when the peritoneal macrophages were depleted. Was this recruitment in mucosal and muscularis layer or just the latter?
- 4) Along the same lines what were these immune cells?
- 5) Along the same lines what chemokines were involved in this recruitment and were the macrophages the source of these chemokines or were they affecting other cells.

Minor

- 1) "They are also known to disappear in the early stage of infection or inflammation^{21, 23, 24} although this 13 may be caused by their aggregatory properties that lead to clump formation rather than actual disappearance." Line 270-271 the second part of this sentence should have a reference perhaps to the Randolph paper in JEM about these cells (ref Zhang N, Czepielewski RS, Jarjour NN, Erlich EC, Esaulova E, Saunders BT, Grover SP, Cleuren AC, Broze GJ, Edelson BT, Mackman N, Zinselmeyer BH, Randolph GJ).
- 2) "Our data indicate that peritoneal cavity macrophages are potential therapeutic targets for numerous intestinal diseases, including IBD." Line 305-306. This last sentence suggests you want to target these macrophages and makes it sound like you want to inhibit them. I imagine you could use better wording to suggest the exact opposite i.e., enhance their repair functions.

Reviewer #2 (Remarks to the Author):

This manuscript follows precedent from Wang and Kubes who have published evidence that Gata6⁺ large peritoneal resident macrophages can invade a site of thermal injury in the liver. Here, the authors have applied the approach to the thermally injured colon. The injury the authors inflicted was on the serosal side of the colon, so it seems rather logical that some cells from the serosal side might attach to the site of injury. The authors attempt to claim that the Gata6⁺ peritoneal macrophages orchestrate response to injury. However, there are several concerns that greatly diminish the credibility of this claim, as follows:

1. In Figure 1, the Gata6 immunostaining appears to be an artifact. Quality Gata6 staining would be nuclear and thus should not overlay with F4/80 staining, which is not nuclear.
2. Given that the authors only used faulty Gata6 staining to identify the peritoneal macs, it would have been helpful to add staining for other markers rather selectively expressed by these macs (ICAM2, CD93, p-selectin?)
3. The authors have no genetic models (such as Gata6 deficient macs) to support their findings.
4. The clodronate loaded liposome approach administered i.p. is not selective enough. Other macrophages, such as those embedded within the serosal cover itself are also depleted by i.p. clodronate.

5. The adoptive transfer experiment lacks controls. The crude peritoneal population was transferred and there was no variations included to be sure that the cells attaching to the wounds were the Gata6+ macs. At the very least, a cohort treated with peritoneal cells with depleted Gata6+ macs would have added to the rigor.

6. Early on in the manuscript, some of the key points used to establish the model and early results were embedded in the supplement. Supplemental figure 2 shows the FACS gating on the macrophages of interest. The gate chosen to identify the resident macrophages is not logical! It's just a gate sitting outside of the CD11b+ area, which catches a minor fraction of cells when there is inflammation. The gate does not capture a distinct population it seems. Also, it is unclear why surface markers enriched on large peritoneal macrophages were not used.

7. Overall, it is expected that a serosal injury would attract peritoneal macrophages on the serosal side of the tissue. The authors claim that the cells "penetrate" into the gut. However, no cross-sections of the gut are shown in the manuscript body to support that conclusion. There is one cross-section shown in Fig S5. The staining for Gata6 was used there and, again, the antibody does not seem specific or localized to where it should be (the nucleus). Thus, the whole concept of macrophages penetrating from the peritoneum to the gut is not really illustrated in the manuscript.

Reviewer #3 (Remarks to the Author):

Review of Honda et al.

The authors describe a very intriguing set of results suggesting that peritoneal macrophages can detect intestinal injury on the luminal side and invade the site of injury for repair. Furthermore, the authors show that the macrophages are not derived systemically through the vasculature; rather, the cells originate from within the peritoneum. The authors also report on a somewhat surprising result that the cells' recruitment occurs independent of the microbiota, even in the context of a colitis model. These would be novel findings, and has potential implications for treating acute injury to the intestine, e.g., using tissue macrophages primed for resolution. Overall, the authors present convincing evidence for the peritoneum as a source of the GATA6+ macrophages. My questions mainly deal with the route of infiltration in the DSS model and other related results discussed in the paper, as the analysis of this injury model was less detailed. On the other hand, one of the main discussion items the is significance of the results in IBD treatment.

1. The authors used an antibiotic cocktail to deplete the intestine of bacteria. I could not find data in the referenced paper confirming the depletion of bacteria. While CFUs from fecal material could be plated, some species may not be culturable, and sequencing may provide a more definitive validation. This would also allow the authors to determine if some selection has occurred.

2. When the authors injected (via IP) the peritoneal cells from LysM-eGFP donor mice in DSS-injured mice, did they observe any effect on body weight, disease activity, colon length, or other histological markers? Based on the CLL depletion experiment results presented by the authors, I would expect the addition of peritoneal cells to provide improved tissue repair.

3. Regarding the DSS colitis model, the authors point to marked inflammatory cell infiltration in CLL treated mice compared to PBS treated mice. Have the authors characterized the inflammatory cell population? Even for the PBS treated animals, which still had DSS-induced injury, I would expect neutrophils in the epithelium and some erosion of crypts. It is difficult to assess the impact of CLL depletion (and hence the importance of the peritoneally derived macrophages) without corresponding data on control animals that have not been exposed to DSS. It would be useful to compare the injury parameters of CLL and PBS treated animals to control animals.

4. An intriguing aspect of this paper was the ATP- and hyaluronan-dependent route of recruitment

for the tissue macrophages. However, this route was described only for the thermal injury model. In the thermal injury model, the authors highlight the accumulation of large macrophages in the muscularis, in contrast to the accumulation of monocytes in the lamina propria. For the DSS model, where the initial site of injury is in the mucosa, it is unclear what the recruitment factors would be and how they would signal the peritoneal macrophages. Have the authors tested if the same factors are involved?

5. Along this vein, have the authors investigated the role of gut bacteria in signaling the peritoneal macrophages in the DSS model? It is well known that DSS treatment results in major alterations of gut microbiota community structure and immunological profile. The discussion, as written, suggests that the microbiome does not affect peritoneal macrophage invasion, but does not specify the injury context.

6. Some mention is warranted in the discussion regarding the distinction between acute injury vs. chronic inflammation. The DSS protocol followed in this paper is a model of acute colitis. It would be interesting to know if the large peritoneal macrophages also play a significant protective role in a chronic inflammation model, which may be more relevant in IBDs such as Crohn's disease. Have the authors monitored the macrophage counts over the 5 day period?

Reviewer #1 (Remarks to the Author):

This study describes a novel concept in the intestinal repair arena, ie., peritoneal macrophages appear to be important in helping repair the intestinal muscularis layer. I have a number of queries.

We thank the reviewer for their positive evaluation of the novelty, interest, and importance of our study. We have gone back to the laboratory and performed all requested experiments.

1) Would DSS colitis cause greater injury in CCR2^{-/-} mice if peritoneal macrophages are depleted?

We performed a preliminary DSS colitis experiment in CCR2^{-/-} mice. The data showed no differences in the body weight change or survival between C57BL/6 and CCR2^{-/-} mice, which suggested CCR2 does not play a pivotal role in this colitis model (**Figure R1**). It is possible that the absence of CCR2 in DSS-induced colitis is compensated for by other receptor/ligand combinations or that DSS might provide such a strong chemical signal that it may directly overcome the effects of CCR2 deficiency.

Figure R1. CCR2 does not play a pivotal role in the 4% DSS-induced colitis model. Body weight change (left) and survival (right) of control and CCR2ko mice with 4% DSS-induced colitis. n = 10–20/group.

2) What would happen if the burn injury occurred from the luminal side of the intestine without breaching the serosal layer? Would the GATA6⁺ macrophages still accumulate or would this be invisible to the peritoneal macrophages?

We thank the reviewer for this suggestion because it explains the mechanisms of GATA6⁺ macrophage recruitment in DSS-induced colitis where inflammation occurs mainly in the mucosal layer. We have imaged the intestinal serosa at 6 h after thermal injury from the mucosal side. Interestingly, peritoneal macrophages sensed the intestinal mucosal injury, although the number was less compared with the injury from the serosal side (**Fig 3g, h**).

Together with the accumulation of peritoneal macrophages in the DSS-induced colitis model, these data suggest that severe intestinal mucosal injury releases a 'find me signal' to the peritoneal cavity.

3) The DSS colitis shows a remarkable increase in immune cells throughout the intestinal layers when the peritoneal macrophages were depleted. Was this recruitment in mucosal and muscularis layer or just the latter?

4) Along the same lines what were these immune cells?

Thank you for these comments. The degrees of infiltration by neutrophils, macrophages, and lymphocytes were compared between PBS-L and CLL groups, but no significant differences were observed in the lamina propria and muscularis, respectively (**Supplementary Fig. 11a, b**). Although the number of neutrophils in the muscularis of the DSS-induced colitis group treated with CLL was approximately twice that of the group treated with PBS-L, it did not reach statistical significance.

5) Along the same lines what chemokines were involved in this recruitment and were the macrophages the source of these chemokines or were they affecting other cells.

We have performed additional DSS-induced colitis experiments and examined chemokine expression in the colon with or without peritoneal macrophage depletion (**Supplementary Fig. 11c**). The data suggest that peritoneal macrophages do not play a pivotal role in the recruitment of other immune cells during systemic intestinal inflammation.

Minor

1) "They are also known to disappear in the early stage of infection or inflammation^{21, 23, 24} although this disappearance may be caused by their aggregatory properties that lead to clump formation rather than actual disappearance." Line 270-271 the second part of this sentence should have a reference perhaps to the Randolph paper in JEM about these cells (ref Zhang N, Czepielewski RS, Jarjour NN, Erlich EC, Esaulova E, Saunders BT, Grover SP, Cleuren AC, Broze GJ, Edelson BT, Mackman N, Zinselmeyer BH, Randolph GJ).

Thank you for this comment. As requested, the reference has been added.

2) "Our data indicate that peritoneal cavity macrophages are potential therapeutic targets for numerous intestinal diseases, including IBD." Line 305-306. This last sentence suggests you want to target these macrophages and makes it sound like you want to inhibit them. I imagine you could use better wording to suggest the exact opposite i.e., enhance their repair functions.

We appreciate this thoughtful suggestion. We rewrote lines 305–306 from “Our data indicate that peritoneal cavity macrophages are potential therapeutic targets for numerous intestinal diseases, including IBD.” to “Our data indicate that enhancing their repair functions is a potential therapeutic approach for numerous intestinal diseases that include IBD.”.

Reviewer #2 (Remarks to the Author):

This manuscript follows precedent from Wang and Kubes who have published evidence that Gata6+ large peritoneal resident macrophages can invade a site of thermal injury in the liver. Here, the authors have applied the approach to the thermally injured colon. The injury the authors inflicted was on the serosal side of the colon, so it seems rather logical that some cells from the serosal side might attach to the site of injury. The authors attempt to claim that the Gata6+ peritoneal macrophages orchestrate response to injury. However, there are several concerns that greatly diminish the credibility of this claim, as follows:

We sincerely thank the reviewer for their constructive detailed evaluation. We have performed the requested experiments as described below. We believe that the manuscript has been greatly improved with more solid and convincing data generated by the additional experiments.

1. In Figure 1, the Gata6 immunostaining appears to be an artifact. Quality Gata6 staining would be nuclear and thus should not overlay with F4/80 staining, which is not nuclear.

Thank you for your valuable suggestions. The appropriate conditions for immunofluorescence staining were reviewed in detail and experiments to validate the identity of F4/80^{hi} macrophages in the injured intestines were redone. We have restained samples carefully and new figures are presented in the revised manuscript with more convincing staining. First, immunohistochemistry (IHC) of formalin-fixed, paraffin-embedded samples of peritoneal macrophages showed that the Gata6 immunostaining does not overlap with F4/80, but it does overlap with nuclear (**Supplementary Fig. 2a**). Similar results were obtained by staining cytospin specimens (**Supplementary Fig. 2b**). Finally, we confirmed that GATA6⁺ peritoneal macrophages accumulate in the injured intestine by IHC under the same conditions (**Fig. 1g, h, Supplementary Fig. 2c, and Supplementary Fig. 8c**).

2. Given that the authors only used faulty Gata6 staining to identify the peritoneal macs, it would have been helpful to add staining for other markers rather selectively expressed by these macs (ICAM2, CD93, p-selectin?)

Thank you for this comment. We have stained peritoneal cells from LysM-eGFP mice using a PE-ICAM2 (CD102) antibody. In total, 78.6% of GFP⁺ peritoneal cells were also positive for CD102, which demonstrated that the double positive cells are large peritoneal macrophages (**Supplementary Fig. 6a–c**). We next transferred peritoneal cells from LysM-eGFP mice to the peritoneal cavity of C57BL/6 mice after staining CD102. Imaging of the intestines at 6 h after injury showed that LysM⁺CD102⁺ cells had accumulated in the injured area, but not in the uninjured area (**Supplementary Fig. 6d–f**). These data strengthened our findings showing that peritoneal cavity macrophages are recruited to injured intestine.

3. The authors have no genetic models (such as Gata6 deficient macs) to support their findings.

Exact genetic models such as GATA6-deficient macrophages may support our findings. However, establishing this genetic model and performing additional experiments is not possible within the time frame provided for the revision with the restrictions of research activities due to the COVID-19 pandemic. However, we have performed many additional experiments to expand our observations and revised the manuscript extensively. Our revised manuscript contains new IHC and flow cytometry data showing that GATA6⁺F4/80^{hi} peritoneal macrophages accumulate in the injured intestine. Moreover, we revealed that peritoneal macrophages positive for ICAM2, which is a specific cell surface marker for large peritoneal macrophages, accumulate within the intestinal injury site. Taken together, these new data strengthen our findings that GATA6⁺ peritoneal cavity macrophages are recruited to injured intestine. Hopefully, we will be able to address this query in the future.

4. The clodronate loaded liposome approach administered i.p. is not selective enough. Other macrophages, such as those embedded within the serosal cover itself are also depleted by i.p. clodronate.

Thank you for these comments. The number of macrophages in the colon was compared between PBS-L and CLL groups in the steady state, but no significant difference was observed in the lamina propria or muscularis, respectively (**Supplementary Fig. 11a, b**). The data show that i.p. administration of CLL does not deplete intestinal macrophages.

We can make this claim in the intestinal burn injury model because of the nature of this injury. We ablate every cell in a 500- μ m area. Therefore, the only way monocytes and macrophages can get there is through recruitment. The baseline at the start of injury in each mouse is zero monocytes, zero vasculature, and zero macrophages. Then, we examined recruited cells in the area. We have highlighted this in the text to avoid suggesting that the

baselines are different.

5. The adoptive transfer experiment lacks controls. The crude peritoneal population was transferred and there was no variations included to be sure that the cells attaching to the wounds were the Gata6+ macs. At the very least, a cohort treated with peritoneal cells with depleted Gata6+ macs would have added to the rigor.

We thank the reviewer for pointing this out. When peritoneal cells from LysM-eGFP mice with depleted GATA6⁺ macrophages by intraperitoneal CLL administration were transferred into C57BL/6 mice intraperitoneally, GFP⁺ cells were not found within the intestinal injury site (Fig. 2k, i). This is now clearly stated in the manuscript.

6. Early on in the manuscript, some of the key points used to establish the model and early results were embedded in the supplement. Supplemental figure 2 shows the FACS gating on the macrophages of interest. The gate chosen to identify the resident macrophages is not logical! It's just a gate sitting outside of the CD11b+ area, which catches a minor fraction of cells when there is inflammation. The gate does not capture a distinct population it seems. Also, it is unclear why surface markers enriched on large peritoneal macrophages were not used.

We agree with the reviewer's opinion. To focus on the milieu of the injury site, we reperformed flow cytometry using cells isolated from injury biopsies at 24 h post-intestinal injury. Reanalysis of our flow cytometry data showed a clear population of GATA6⁺CD11b^{hi}F4/80^{hi} macrophages in the injured colon. We have amended the figure accordingly (new **Supplementary Fig. 3**).

7. Overall, it is expected that a serosal injury would attract peritoneal macrophages on the serosal side of the tissue. The authors claim that the cells "penetrate" into the gut. However, no cross-sections of the gut are shown in the manuscript body to support that conclusion. There is one cross-section shown in Fig S5. The staining for Gata6 was used there and, again, the antibody does not seem specific or localized to where it should be (the nucleus). Thus, the whole concept of macrophages penetrating from the peritoneum to the gut is not really illustrated in the manuscript.

As described, we have reperformed the IFC experiments and confirmed that GATA6⁺ peritoneal macrophages accumulate in the injured intestine (Fig. 1g, h, **Supplementary Fig. 2c**, and **Supplementary Fig. 8c**). Indeed, some GATA6⁺ peritoneal macrophages appeared to penetrate into the deeper area of the gut.

Many GATA6⁺ macrophages remained on the surface of the injury site in the newly

prepared immunostained specimens, and therefore we avoided the word “penetrate” in the revised manuscript and instead mainly used the word “accumulate”. However, it cannot be ruled out that the peritoneal macrophages that had invaded the injured site might lose expression of GATA6, which would make it difficult to identify macrophages derived from the peritoneal cavity in the deep area of the intestinal injury. In any case, our dataset manuscript clearly shows that GATA6⁺ peritoneal macrophages rapidly accumulate at the site of intestinal injury from the serosal site and facilitate tissue repair.

Reviewer #3 (Remarks to the Author):

Review of Honda et al.

The authors describe a very intriguing set of results suggesting that peritoneal macrophages can detect intestinal injury on the luminal side and invade the site of injury for repair. Furthermore, the authors show that the macrophages are not derived systemically through the vasculature; rather, the cells originate from within the peritoneum. The authors also report on a somewhat surprising result that the cells' recruitment occurs independent of the microbiota, even in the context of a colitis model. These would be novel findings, and has potential implications for treating acute injury to the intestine, e.g., using tissue macrophages primed for resolution. Overall, the authors present convincing evidence for the peritoneum as a source of the GATA6⁺ macrophages. My questions mainly deal with the route of infiltration in the DSS model and other related results discussed in the paper, as the analysis of this injury model was less detailed. On the other hand, one of the main discussion items the is significance of the results in IBD treatment.

We sincerely thank the reviewer for their positive evaluation of the novelty, interest, and importance of our study. We have gone back to the laboratory and performed all requested experiments.

1. The authors used an antibiotic cocktail to deplete the intestine of bacteria. I could not find data in the referenced paper confirming the depletion of bacteria. While CFUs from fecal material could be plated, some species may not be culturable, and sequencing may provide a more definitive validation. This would also allow the authors to determine if some selection has occurred.

The SYTOX method has been used to detect DNA of dead bacteria in feces. The protocol has been established (Macpherson et al., Encyclopedia of Microbiology 2009) and is accurate to examine the change in bacterial load caused by Abx treatment. Indeed, we have

previously shown that 99.6% of bacterial DNA in feces is diminished by Abx (Honda et al., Nat Commun 2020, Supplementary Fig. 4a, doi: 10.1038/s41467-020-15068-4).

We performed sequencing analyses of feces, which showed differences of microbial communities between control and Abx-treated mice (**Supplementary Fig. 7a–d**). For example, the proportions of phyla *Firmicutes* and *Bacteroidetes* were significantly lower in Abx-treated mice (**Fig. 4c**). Although these sequence data may not be directly related to the manuscript, the data provide a more definitive validation of the influence of Abx on gut microbiota. Thank you for pointing this out.

2. When the authors injected (via IP) the peritoneal cells from LysM-eGFP donor mice in DSS-injured mice, did they observe any effect on body weight, disease activity, colon length, or other histological markers? Based on the CLL depletion experiment results presented by the authors, I would expect the addition of peritoneal cells to provide improved tissue repair. As described in the discussion, Seo et al. have shown that adoptive i.p. transfer of peritoneal macrophages with IL-33 ameliorates inflammation in a preclinical colitis model (Sci Rep 7, 851 (2017)). Furthermore, we conducted additional peritoneal cell (PC) transfer experiments in DSS-induced colitis (1.5×10^6 PC i.p. transfer at 0, 2, and 4 days after the start of 4% DSS-containing water) and evaluated the body weight change, disease activity index, colon length, and histomorphological scores (**Figure 7f–k**). In comparison with the PBS-treated control group, the PC transfer group had a lower disease activity index, while the other parameters did not show statistically significant differences. Although we cannot definitively show the best treatment strategy using PC transfer method for DSS-induced colitis, we believe that a more detailed examination would be beyond the focus of our manuscript and should be planned for a subsequent investigation in the future.

3. Regarding the DSS colitis model, the authors point to marked inflammatory cell infiltration in CLL treated mice compared to PBS treated mice. Have the authors characterized the inflammatory cell population? Even for the PBS treated animals, which still had DSS-induced injury, I would expect neutrophils in the epithelium and some erosion of crypts. It is difficult to assess the impact of CLL depletion (and hence the importance of the peritoneally derived macrophages) without corresponding data on control animals that have not been exposed to DSS. It would be useful to compare the injury parameters of CLL and PBS treated animals to control animals.

The degrees of infiltration by neutrophils, macrophages, and lymphocytes were compared between PBS-L and CLL groups, but no significant difference was observed in the lamina propria or muscularis, respectively (**Supplementary Fig. 11a, b**). Although the number of

neutrophils in the muscularis of the DSS-induced colitis group treated with CLL was approximately twice that of the group treated with PBS-L, it did not reach statistical significance.

In response to the second question, we have added appropriate controls to DSS experiments. CLL administration did not have a significant influence on mice that were not exposed to DSS-containing water.

4. An intriguing aspect of this paper was the ATP- and hyaluronan-dependent route of recruitment for the tissue macrophages. However, this route was described only for the thermal injury model. In the thermal injury model, the authors highlight the accumulation of large macrophages in the muscularis, in contrast to the accumulation of monocytes in the lamina propria. For the DSS model, where the initial site of injury is in the mucosa, it is unclear what the recruitment factors would be and how they would signal the peritoneal macrophages. Have the authors tested if the same factors are involved?

Thank you for this suggestion. We have imaged the intestinal serosa at 6 h after thermal injury from the mucosal side. Interestingly, peritoneal macrophages sensed the intestinal injury from the mucosal side, although the number was less compared with the injury from the serosal side (**Fig 3g, h**). Together with the accumulation of peritoneal macrophages in the DSS colitis model, these data suggest that severe intestinal mucosal injury releases a 'find me signal' to the peritoneal cavity.

We also investigated the HABP-positive area in the DSS-induced colitis model. Immunofluorescence staining showed increased expression of hyaluronan, which is the ligand for CD44, in the colon of DSS-induced colitis compared with the control (**Fig 6g, h**). This demonstrates a specific causal link between large peritoneal macrophages and the inflamed colon.

5. Along this vein, have the authors investigated the role of gut bacteria in signaling the peritoneal macrophages in the DSS model? It is well known that DSS treatment results in major alterations of gut microbiota community structure and immunological profile. The discussion, as written, suggests that the microbiome does not affect peritoneal macrophage invasion, but does not specify the injury context.

We performed additional experiments and examined recruitment of peritoneal macrophages in control and Abx-treated mice at 5 days after the start of 4% DSS-containing water. The data reveal that Abx treatment did not influence the recruitment of F4/80^{hi} large peritoneal macrophages to the colon of the DSS-induced colitis model. These new data are presented in **Figure 6i, j** and described in the manuscript. Thank you for this important comment.

6. Some mention is warranted in the discussion regarding the distinction between acute injury vs. chronic inflammation. The DSS protocol followed in this paper is a model of acute colitis. It would be interesting to know if the large peritoneal macrophages also play a significant protective role in a chronic inflammation model, which may be more relevant in IBDs such as Crohn's disease. Have the authors monitored the macrophage counts over the 5 day period?

Thank you for these comments. We have performed additional experiments and examined the accumulation of peritoneal macrophages in more chronic colitis models using 2% DSS-containing water for 7 days. Imaging showed similar accumulation of peritoneal macrophages in the colon of a chronic DSS-induced colitis model (**Supplementary Fig. 10a–c**).

REVIEWERS' COMMENTS

Reviewer #1 (Remarks to the Author):

Your rebuttal is acceptable and well done.

Reviewer #3 (Remarks to the Author):

Review of Honda et al.

The authors have thoughtfully addressed my major comments. I especially appreciate the additional experiments included in the revision, e.g., effect of injury from mucosal side and lack of Abx impact on peritoneal macrophage recruitment in DSS model. I do not have further comments.

Reviewer #1 (Remarks to the Author):

Your rebuttal is acceptable and well done.

We sincerely thank the reviewer's positive evaluation of the novelty, interest, and importance of our revised manuscript.

Reviewer #3 (Remarks to the Author):

Review of Honda et al.

The authors have thoughtfully addressed my major comments. I especially appreciate the additional experiments included in the revision, e.g., effect of injury from mucosal side and lack of Abx impact on peritoneal macrophage recruitment in DSS model. I do not have further comments.

We sincerely thank the reviewer's positive evaluation of the novelty, interest, and importance of our revised manuscript.